# Optimizing Capacitive Pressure Sensor Geometry: A Design of Experiments Approach with a Computer-Generated Model

**DOI:** 10.3390/s24113504

**Published:** 2024-05-29

**Authors:** Kiran Keshyagol, Shivashankarayya Hiremath, Vishwanatha H. M., Achutha Kini U., Nithesh Naik, Pavan Hiremath

**Affiliations:** 1Department of Mechatronics, Manipal Institute of Technology, Manipal Academy of Higher Education, Manipal 576104, India; kiran2.mitmpl2022@learner.manipal.edu (K.K.); ss.hiremath@manipal.edu (S.H.); 2Survivability Signal Intelligence Research Center, Hanyang University, Seongdong-gu, Seoul 04763, Republic of Korea; 3Department of Mechanical and Industrial Engineering, Manipal Institute of Technology, Manipal Academy of Higher Education, Manipal 576104, India; vishwanatha.hm@manipal.edu (V.H.M.); achutha.kini@manipal.edu (A.K.U.); nithesh.naik@manipal.edu (N.N.)

**Keywords:** capacitive pressure sensor, sensitivity, dielectrics, optimization, design of experiment, PDMS, PVDF

## Abstract

This study presents a comprehensive investigation into the design and optimization of capacitive pressure sensors (CPSs) for their integration into capacitive touch buttons in electronic applications. Using the Finite Element Method (FEM), various geometries of dielectric layers were meticulously modeled and analyzed for their capacitive and sensitivity parameters. The flexible elastomer polydimethylsiloxane (PDMS) is used as a diaphragm, and polyvinylidene fluoride (PVDF) is a flexible material that acts as a dielectric medium. The Design of Experiment (DoE) techniques, aided by statistical analysis, were employed to identify the optimal geometric shapes of the CPS model. From the prediction using the DoE approach, it is observed that the cylindrical-shaped dielectric medium has better sensitivity. Using this optimal configuration, the CPS was further examined across a range of dielectric layer thicknesses to determine the capacitance, stored electrical energy, displacement, and stress levels at uniform pressures ranging from 0 to 200 kPa. Employing a 0.1 mm dielectric layer thickness yields heightened sensitivity and capacitance values, which is consistent with theoretical efforts. At a pressure of 200 kPa, the sensor achieves a maximum capacitance of 33.3 pF, with a total stored electric energy of 15.9 × 10^−12^ J and 0.468 pF/Pa of sensitivity for 0.1 dielectric thickness. These findings underscore the efficacy of the proposed CPS model for integration into capacitive touch buttons in electronic devices and e-skin applications, thereby offering promising advancements in sensor technology.

## 1. Introduction

In the modern era, the pursuit of comfort and efficiency has spurred the integration of cutting-edge technologies, such as the Internet of Things, robotics, wearable devices, and pressure sensors, into everyday life. Among these innovations, capacitive pressure sensors have emerged as indispensable components in a multitude of industries, including automotive, aerospace, healthcare, and consumer electronics [1]. By converting pressure variations into electrical signals, these sensors enable precise measurements and the control of pressure-sensitive processes, thereby playing a crucial role in enhancing operational efficiency and ensuring product reliability [2]. However, the effectiveness of capacitive pressure sensors depends heavily on their design, particularly on the intricate geometry of their sensing elements. It is imperative to optimize the geometry of CPS structures to enhance their performance characteristics, including their sensitivity, linearity, and dynamic range [3].

Traditional methods of optimization often involve arduous trial-and-error processes, which are not only time-consuming and resource-intensive, but also may not yield optimal results. To overcome these challenges, the DoE methodology presents itself as a systematic and efficient approach in optimizing sensor geometry while minimizing the number of experiments required. The application of DoE enables us to comprehensively explore the design space and unravel the intricate interaction effects between the different parameters. Through this rigorous exploration, the optimal combination of geometric attributes that bestow the desired sensor characteristics can be identified, thus paving the way for significant advancements in CPS design. Furthermore, as materials for these sensors exhibit a broad spectrum of mechanical and electrical parameters, it is imperative to tailor their design to suit specific application requirements. PDMS is a silicone elastomer material that is known for its remarkable flexibility and biocompatibility [4]. Its excellent mechanical properties include low surface tension and high elasticity. It was selected to model the diaphragm of the sensor owing to its advantageous mechanical properties [5]. Conversely, PVDF, which is a thermoplastic polymer, exhibits distinctive piezoelectric and pyroelectric characteristics [6]. These unique properties make PVDF an ideal material for applications such as sensors, actuators, and energy-harvesting devices [7,8]. Therefore, PVDF was used as the dielectric material in this study.

This study aims to investigate the impact of varying the dielectric geometry on the performance of CPS. It explores the adoption of different combinations of dielectric geometries, diaphragm shapes, and mesh densities to optimize the sensitivity and reliability of CPS. Using the DoE approach, the most effective combinations exhibited high desirability in terms of sensitivity. Furthermore, the electrical parameters of the CPS were analyzed to understand the relationship between pressure and capacitance. Additionally, the mechanical parameters of the CPS, including deflection and stress distribution, were investigated under varying pressure conditions. Thus, this study contributes to the optimization and performance analysis of CPS by emphasizing the importance of dielectric geometry in enhancing sensitivity and reliability. The insights gained from this study can inform the development of more efficient and accurate capacitive pressure sensors for various applications.

## 2. Methodology

The proposed CPS comprises a four-layered structure, as illustrated in Figure 1. The top layer is a diaphragm made of flexible materials such as PDMS. The second and fourth layers consist of thin parallel plate electrodes that serve as power-supply interfaces for the CPS and that are usually made up of highly electrically conductive materials such as gold, silver, and copper. The third layer was the intermediate layer, positioned between the two electrodes [9]. This dielectric layer exhibited compressibility characteristics that responded to the pressure applied to the diaphragm. Under uniform pressure, the diaphragm undergoes compression, which causes a variation in the thickness of the flexible dielectric layer [10]. This change in thickness directly influences the overall capacitance of the CPS. Consequently, the capacitive response of the sensor is modulated by the mechanical deformation induced by the applied pressure on the diaphragm [11].

### 2.1. Change in the Capacitance under Compression

When pressure is exerted on the diaphragm, it undergoes compression, leading to a change in its shape and, consequently, altering the distance between two electrodes, affecting the capacitance of the CPS. The change in capacitance, resulting from this phenomenon, can be quantified by Equation (1):(1)$$\Delta C=C-C_{0}=\varepsilon A\frac{\Delta d}{d\left(d-\Delta d\right)}$$
where $C_{0}$ represents the capacitance without pressure, ∆C is the variation in capacitance owing to changes in the dielectric thickness, ‘A’ denotes the contact area of the electrode, ε denotes the dielectric constant of the material situated between the electrodes, ‘d’ is the thickness of the dielectric medium without pressure, and Δd represents the change in distance between the two electrodes caused by the applied pressure [12].

### 2.2. Displacement of the Diaphragm

Different geometric shapes can be chosen based on specific requirements; however, geometry plays a crucial role in determining the mechanical parameters, such as diaphragm displacement, and consequently, stress levels.

The authors of the work presented in [13] fabricated circular-shaped sensors using PDMS with different concentrations of a curing agent, resulting in the highest sensitivity of a wide pressure range with the pressure sensitivity in the range of 0.3 per kPa to 3.2 per MPa. Researchers opted to use square-shaped PDMS to obtain a higher-pressure sensitivity of 0.023 kPa^−1^ [14]. These shapes behaved similarly to those of the applied pressure [15]. The central deflection (w) of a flat and circular diaphragm with clamped edges subjected to a uniform pressure is given by Equation (2).
(2)$$w\left(r\right)=\frac{P\left(R^{2}-r^{2}\right)}{64D}$$
where w is the deflection, r is the radial distance from the center of the diaphragm, R is the radius of the diaphragm, P is the applied pressure, and D is the flexural rigidity (Equation (3)) [16].
(3)$$D=\frac{Eh^{3}}{12\left(1-v^{2}\right)}$$

In this case, E represents the Young’s modulus, h represents the thickness, and v denotes the Poisson’s ratio of the diaphragm material.

### 2.3. Stress Level upon External Load

The von Mises stress graph is a common tool in solid mechanics which is used to analyze stress distribution within a material under external loading. It provides crucial information about the permissible stress on a diaphragm [12]. Von Mises stress is a measure that combines the effects of normal and shear stresses in a material. In a typical 2D stress state, represented in Figure 2a, the independent stress components are illustrated as *σ*_x_ and *σ*_y_, while a third stress component, *σ*_3_, may exist along the *z*-axis. When dealing with complex stress systems, the initial step involves determining the principal stresses *σ*_x_, *σ*_y_, and *σ*_z_. From these, three key stress indicators are derived: principal stress, maximum shear stress, and von Mises stress [17].
(4)$$\sigma_{1},\;\sigma_{2}=\frac{\sigma_{x}+\sigma_{y}}{2}\pm \sqrt{\left(\frac{\sigma_{x}-\sigma_{y}}{2}\right)}+\tau_{xy}^{2}$$
where $\sigma_{3}=0$.

The maximum shear stress is calculated using Equations (5)–(7).
(5)$$\tau_{max,1,2}=\frac{\sigma_{1}+\sigma_{2}}{2}$$
(6)$$\tau_{max,1,3}=\frac{\sigma_{1}+\sigma_{3}}{2}$$
(7)$$\tau_{max,2,3}=\frac{\sigma_{2}+\sigma_{3}}{2}$$

von Mises stress in terms of principal stress is shown in Equation (8).
(8)$$\sigma_{v}=\sqrt{\frac{{{(\sigma }_{1}-\sigma_{2})}^{2}+{{(\sigma }_{2}-\sigma_{3})}^{2}+{{(\sigma }_{1}-\sigma_{3})}^{2}}{2}}$$

When, $\sigma_{3}=0$, Equation (8) becomes,
(9)$$\sigma_{v}=\sqrt{\sigma_{1}^{2}+\sigma_{2}^{2}-\sigma_{1}\sigma_{2}}$$

By considering only $\sigma_{x}$ and $\tau_{xy}$, the von Misses stress equation can be shown as Equation (10).
(10)$$\sigma_{v}=\sqrt{\sigma_{x}^{2}+3\tau_{xy}^{2}}$$

### 2.4. Energy Storage in the Capacitive Model

The region influenced by the electric field between the two electrodes was delineated using the concept of capacitance [18]. An electric field emerges between the two metallic plates when a potential difference is applied across them, if there is a separation between the plates, as shown in Figure 2b. The equation for the electrostatic energy (U) of the continuous charge distribution, which forms the electric field (E) in each volume (v), is given as:(11)$$U=\frac{1}{2}\varepsilon \int_{v}{\left|E\right|}^{2}dv$$
where dv is the differential volume element and ε is the permittivity of the medium in which the field exists.

### 2.5. Sensitivity

The sensitivity of a sensor refers to its capacity to detect small changes in the input signal, resulting in significant variations in the output signal. As highlighted in the work presented in [19], the sensitivity of a capacitor sensor (S) is defined as the change in capacitance (ΔC) per unit change in pressure (ΔP). This relationship is typically represented by Equation (12).
(12)$$s=\frac{d(∆C)}{d(∆P)}$$

### 2.6. FE Analysis

Boundary conditions play a crucial role in FEM analysis. In this study, all materials are considered to be solid, homogeneous, and linearly elastic. It is assumed that there are no initial stresses or displacements. Fixed constraints are imposed at the edges of the diaphragm, positive electrode, and complete ground electrode, while the dielectric columns are left unconstrained. Following this, a boundary load is applied to the diaphragm’s surface. The entire CPS is initially assigned zero electric potential and zero electric charges. To facilitate further analysis, a potential difference of one volt is applied between the two electrodes. Because the length of the electrode is far greater than the thickness of the dielectric, the fringing effect of the capacitance is neglected. The contact area between the dielectric geometry and the electrode was approximately the same for all dielectric geometries.

### 2.7. Material Properties and Dimensions

The properties of the proposed model materials are listed in Table 1. PDMS has been widely utilized as a flexible dielectric material [20] and diaphragm in capacitive pressure sensors, owing to its favorable characteristics, including low Young’s modulus, flexibility, biocompatibility, and excellent dielectric properties [21,22]. Silver is used as the electrode material in most capacitive applications [23,24,25,26]. Moreover, owing to its high dielectric value and mechanical flexibility, PVDF is employed as a dielectric material [27]. The geometry of the diaphragm varies between square and circular shapes while maintaining approximately equal overall diaphragm areas. Similarly, the geometry of the dielectric medium was altered while ensuring that the contact area of the electrode remained consistent across all dielectric shapes.

### 2.8. Geometry of the Diaphragm

The diaphragm geometry is a crucial parameter that influences the touch contact area, which directly affects the CPS sensitivity. Different shapes, such as square, circular, or even irregular shapes, can be employed based on the specific requirements of the application. The diaphragm geometry affects the sensitivity, response time, and mechanical stability of the capacitive sensor. Therefore, selecting an appropriate diaphragm shape is essential to optimize the overall performance of the sensor system [15,28]. In this analysis, square and circular geometries were considered to understand the CPS performance. For physical compatibility, electrodes with the same geometry were used. The diaphragm and electrode assembly for both shapes are shown in Figure 3a,f, respectively. Their dimensions are listed in Table 2.

### 2.9. Geometry of the Dielectric Medium

Various geometries have been explored for the dielectric layer in the design of capacitive pressure sensors. Four distinct shapes are considered: conical, cut-spherical, cubical, and cylindrical. Each geometry has unique characteristics that can influence the sensor’s performance. The conical shape featured a tapered structure with a pointed tip, resembling a cone. The cut-spherical geometry resembles a spherical shape, with a portion sliced off to create a flat surface. The cubical shape had a simple box-like structure with straight edges and flat faces. Finally, the cylindrical geometry features a circular cross section with a uniform diameter along its thickness. These geometries are visualized and illustrated in Figure 3 for reference purposes. Additionally, the physical dimensions of each geometry, including the height, diameter, and surface area, are tabulated in Table 3. This comprehensive approach allowed for a systematic evaluation of the different geometries and their suitability for use in the CPS designs.

The geometry of the dielectric medium of a CPS plays a crucial role in determining its performance characteristics. The geometry of the dielectric material, which is situated between the electrodes, and the shape, size, and dimensions of the dielectric material affect the sensor’s capacitance, mechanical flexibility, and overall sensitivity to pressure variations.

In this study, these factors were systematically examined using several combinations of dielectric geometries and different diaphragm shapes. These combinations were evaluated to determine their effectiveness in optimizing the CPS design. The variation in the dielectric geometry with different diaphragm configurations is shown in Figure 3. Three mesh densities were considered for the present FEM analysis. For the normal mesh, element sizes ranging from 0.1 to 1.5 mm were considered (Figure 4a). For the fine mesh, element sizes ranging from 0.1 to 0.8 mm were utilized (Figure 4b). Finally, for the finer mesh, element sizes ranging from 0.04 to 0.55 mm were employed (Figure 4c).

### 2.10. Optimization of the CPS Geometry Using DoE

In this section, the optimization of the sensor geometry is considered, and a square and circular diaphragm and electrode setup are explored. The sensor dimensions, such as the length, width, thickness, and gap between the plates, were maintained approximately constant for all geometries. The sensor operates under a uniform external pressure, and its performance depends primarily on the geometries of the diaphragm, electrode, and dielectric layer. In simulation-based modeling, the type of mesh density is an influential factor. In this study, three mesh densities were used for optimization. Moreover, the gap between the two electrodes was established to have a linear effect on capacitance. This work delves into discerning the individual effects of sensor geometries on sensor performance.

The DoE is meticulously crafted and executed on sensor geometries by employing a 4 × 3 × 2 factorial with randomness of combination of an orthogonal array [29]. Furthermore, ANOVA was applied to the sensor, providing a comprehensive insight into the effect analysis of individual sensor geometries. This study specifically focuses on electrode geometry, dielectric geometry, and mesh density as factors, with sensitivity serving as the response variable. The objective was to ascertain the optimal combination of the aforementioned factors to achieve maximum sensitivity. Detailed descriptions of these three factors and their respective levels are provided in Table 4. This experiment had three factor levels. Factor X had three levels, Y had four levels, and factor Z had two levels. The mesh density was assigned to factor X, the dielectric geometry was assigned to Y, and the dielectric layer geometry was assigned to factor Z.

In this study, all the factors were considered categorical data types, therefore, the classical models were not suitable. The most suitable model for this case is a customized or a computer-generated model. The first-order model used to screen the factors is shown in Equation (13), and the regression model for the main effects is given by Equation (14).
y = b_0_ + b_1_X +b_2_Y + b_3_Z+ ε(13)

The first-order model with interaction is shown in the equation below.
y = b_0_ + b_1_X +b_2_Y + b_3_Z+ b_12_XY+ b_23_YZ + b_31_ZX+ ε(14)
where y is the predicted response, bi is the coefficient of the equation, and ε is uncontrolled noise. Screening was initially performed to identify factors influencing the main effect. It is observed that the factors Y, Y × Z, and Z have the main effects. By screening, it is evident that the Factor X individual, X × Z, and Z individual interactions have no significant effect on the response, therefore, these can be neglected from the regression model, and it is also observed that there is no quadratic effect. The computer-generated model provided all possible combinations of 28 runs. Simulation results are obtained for all combinations; therefore, it is assumed that the same results are obtained for every run with the same settings. Hence, no replication of the treatment is considered, as can be seen in Table 5.

## 3. Result and Discussion

### 3.1. Analysis of Variance (ANOVA)

Table 6 shows the results of the linear regression analysis for the present optimization. The source column indicates the source of the variation in the analysis. ‘Model’ represents the variability explained by the regression model, while ‘Error’ represents the unexplained variability. The F-ratio is calculated by dividing the mean square of the model by the mean square of the error. It represents the ratio of the explained variance to the unexplained variance. In this case, for the ‘Model’, it is 13.6232. The ANOVA results suggest that the regression model is highly significant in explaining variability in the response variable. An extremely low *p*-value (<0.0001) indicates strong evidence against the null hypothesis, implying that at least one predictor variable in the model has a significant effect on the response variable.

Figure 5 shows the response surface methodology (RSM) optimization plot for the sensitivity of the CPS. This plot helps to determine the best combination of parameters that will result in a higher sensitivity within the range. The optimal parameter values are indicated in red in the figure. By using these values for FEA modeling, desirable sensitivity can be achieved. In this case, the expected desirability predicted is approximately 90.5%. The best combination of parameter levels to obtain a higher sensitivity is a circular geometry for the diaphragm and electrodes and a cylindrical geometry for the dielectric structure. For a better FEM, a finer mesh density is the most desirable.

### 3.2. Electrical Parameter Analysis

For the optimized settings, 90.5% desirability of the sensitivity of the CPS, the capacitance was measured by applying a pressure in the range of 0–200 kPa on the surface of the diaphragm, which causes a displacement proportional to the applied pressure. This displacement compresses the dielectric layer, consequently reducing the distance between the two electrodes. Figure 6a–c illustrates the displacement of the diaphragm and the electrical potential distribution in the CPS owing to the three different applied pressure levels. Figure 6d–f shows that the electrical potential is significantly distributed throughout the electrostatic area and is more concentrated at the center owing to the compression of the diaphragm, leading to an increase in capacitance. The electrode length was significantly greater than the thickness of the dielectric layer, therefore, the fringing effect was neglected. Figure 7a shows plots of the capacitance and the total energy stored in the CPS owing to the variation in the applied external pressure, which linearly increases with pressure. In this case, a maximum capacitance of 33.3 and 16.13 pF with a total stored electric energy of 15.9 × 10^−12^ and 7.87 × 10^−12^ J for 0.1 and 0.2 mm thickness of dielectric layer, respectively.

### 3.3. Mechanical Parameter Analysis

The initial phase of the study involved the simplification and execution of multiphysics simulations of the proposed CPS diaphragm. The simulation examined the central deflection of the diaphragm in response to varying external pressures. Observations revealed that the deflection was uniformly distributed across the center of the diaphragm owing to the pressure applied on its surface. The findings indicated a linear increase in deflection with small increments in pressure. Specifically, at 200 kPa, the CPS displacement observed as 33.33 µm and 34.3 µm for 0.1 mm and 0.2-mm thickness of dielectric, respectively. The performance analysis of the PDMS diaphragm suggested a consistent linear relationship between pressure and deflection. Furthermore, the simulation results depicted in Figure 7c highlight the significant impact of pressure on the deflection in the diaphragm and dielectric layers. Figure 7d provides a visual representation of the maximum von Mises stress distribution on the CPS diaphragm for applied pressure and elastic deformation on the diaphragm surface. As the pressure increased, the maximum von Mises stress increased linearly in the diaphragm material. This behavior suggests that the distribution of von Mises stress on the diaphragm increases linearly, implying the generation of thermal energy within the materials. Therefore, the analysis of the thermal effect on the performance of CPS becomes important.

### 3.4. Sensitivity

The dielectric layer thickness is crucial for the capacitance and hence for the sensitivity of the sensor. The relationship between the thickness of the dielectric layer, the capacitance, and the sensitivity is shown in Figure 7b. A higher sensitivity of 0.468 pF/Pa for 0.1 dielectric thickness is obtained. For 0.2 mm dielectric thickness, the sensitivity drastically decreases to 0.112 pF/Pa.

### 3.5. Temperature Effect on Capacitance of CPS

The materials used in the CPS, such as PDMS, PVDF, and Silver, have different coefficients of thermal expansion. As the temperature rises, these materials expand at different rates, leading to mechanical stress within the sensor structure. This stress alters the geometry of the sensor, affecting the distance between the capacitor electrodes and thus the capacitance. In the present study varying applied pressures from 0 to 200 kPa and operating temperatures spanning 20–50 °C are examined. The contour plots in Figure 8a–c depict CPS stress resulting from applied pressures of 20, 100, and 200 kPa on the diaphragm at 20 °C reference temperature. Figure 8d–f illustrate stress development at an operating temperature of 30 °C and pressures of 20, 100, and 200 kPa respectively. The dielectric constant of materials used in the CPS can change with temperature. Typically, as temperature increases, the dielectric constant of materials like PVDF decreases. Since capacitance is directly proportional to the dielectric constant, a decrease in dielectric constant leads to a reduction in capacitance. The effect temperature on the capacitance model is depicted in Figure 9. An increase in temperature results in a higher leakage current between capacitor plates. This leakage current can reduce the effective capacitance of the sensor. These effects combine to result in a decrease in capacitance as the operating temperature of the CPS rises. This reduction in capacitance can impact the sensor’s performance and accuracy, making it important to account for temperature effects in CPS design and calibration.

### 3.6. Comparative Study of Mesh Optimization

In this section, all parameters are studied for different mesh densities, evaluated, and compared to the optimized mesh density obtained from the DoE analysis. In Figure 10a,b, it is evident that mesh density affects the electrical and mechanical parameters in the FEM analysis [30]. The normal mesh density configuration exhibits lower capacitance values, consequently leading to decreased total electrical energy stored in the CPS. Notably, a finer mesh density gives an approximately equal capacitance as that of the analytical capacitance value, hence following the same trend in the total energy stored in the CPS. Regarding the mechanical parameters (Figure 10c,d), for displacement, all three plots overlapped, showing the same effects. All mesh densities show a negligible effect on von Mises stress at lower pressure levels (up to 50 kPa). After 50 kPa, all the mesh densities show a significant change in the values. The plot for finer mesh density shows approximately the same yielding value. Hence, perhaps for this reason, the results of the design of the experimental statistical method predicted a finer mesh density as the optimum size.

### 3.7. A Comparative Study on the Existing Geometry of Capacitive Sensors

This section provides insights into the diverse array of materials and structures utilized in recent studies, along with their corresponding sensitivity characteristics. The reviewed studies encompass various materials and structures. Dielectric materials include traditional polymers, such as PDMS, as well as innovative materials, such as porous eco-flex, rose petal-PMMA double-layer PDMS, and bamboo leaves. Electrode materials range from conventional metals such as gold and aluminum to conductive fabrics, silver nanowires, and spiky nickel. Additionally, some studies have explored composite materials, such as PDMS scrubber composites and PVA/H_3_PO_4_ ionic films.

The details of existing research on the different geometries of the dielectric layer utilized are given in Table 7. The sensor structures exhibit significant diversity, including flat squares, microarrays, 3D structures, cone shapes, and textile arrays. Microstructures, porous configurations, and elastomeric materials are also prevalent, offering advantages, such as enhanced sensitivity and flexibility. Sensitivity, quantified in terms of the pressure-to-capacitance ratio or pressure-induced capacitance change, varied widely among the reviewed studies. Sensitivities range from ultralow values of 0.0040 kPa^−1^ to exceptionally high values exceeding 200 kPa^−1^, depending on the specific material, structure, and fabrication technique.

The versatile application of capacitive pressure sensors encompasses fields such as e-skin, wearable devices, and biomedical tools, highlighting their pivotal role in modern technological advancements. Employing materials like PDMS, PVDF, or their composite forms with elastomers and metal oxide fillers significantly augments sensor flexibility and performance. Metal oxides play an important role in elevating dielectric permittivity, thereby enhancing capacitance. Comparative analysis against existing sensor data, as presented in Table 7, underscores the exceptional performance of the proposed sensor geometry, particularly with a 0.1 mm dielectric layer thickness. This configuration not only exhibits superior sensitivity compared to counterparts in the table but also aligns closely with theoretical capacitance predictions. Notably, achieving a peak capacitance of 33.3 pF pF under a pressure of 200 kPa underscores its efficacy in detecting and responding to pressure fluctuations accurately. The selection of PVDF as the dielectric material stands out as a significant contributor to the heightened capacitance due to its higher dielectric permittivity. Furthermore, the sensor’s ability to achieve a maximum displacement of 33.3 µm further supports its suitability for applications necessitating precision in measurements. This amalgamation of material selection and geometric design establishes the proposed sensor as a frontrunner in the domain, promising enhanced performance and reliability across various practical applications.

## 4. Conclusions

Current research has delved deeply into optimizing and analyzing the performance of CPS by varying the geometry of the dielectric layer. This study underscores the significance of optimizing both the electrode and dielectric geometries, along with mesh density, to improve the CPS performance. By employing the DoE approach, optimized combinations were identified, demonstrating high desirability in terms of sensitivity, thereby validating the efficacy of the optimization strategy. Moreover, the investigation of electrical parameters revealed a linear correlation between pressure and capacitance, indicating that the capacitance increases proportionally with the applied pressure. This finding highlights the suitability of CPS for accurately measuring pressure variations within a specified range. Furthermore, the analysis of the mechanical parameters showed a consistent linear increase in deflection with incremental pressure, indicating the responsiveness and reliability of CPS in detecting pressure changes. Additionally, the von Mises stress distribution was examined to gain insights into the mechanical behavior of CPS under varying pressure conditions. Sensitivity analysis indicated the potential of CPS to achieve high sensitivities, particularly with the optimized dielectric-layer thicknesses. The observed linear relationship between the sensitivity and applied pressure further underscores the robustness of the CPS in pressure-sensing applications.

## Figures and Tables

**Figure 1 sensors-24-03504-f001:**
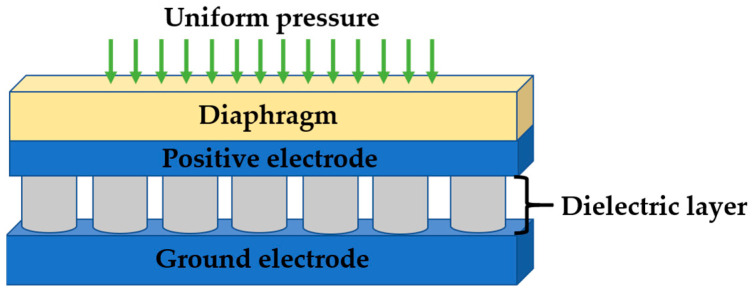
Schematic diagram of the capacitive pressure sensor.

**Figure 2 sensors-24-03504-f002:**
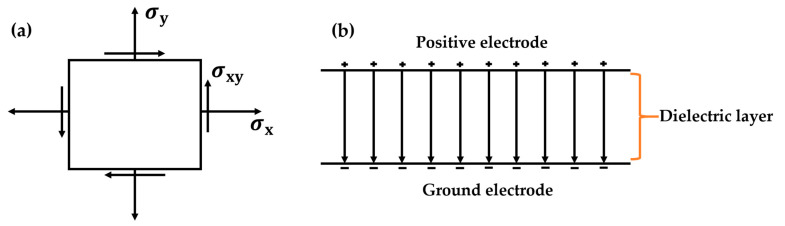
(**a**) 2D stress components, (**b**) The electric field in the dielectric layer.

**Figure 3 sensors-24-03504-f003:**
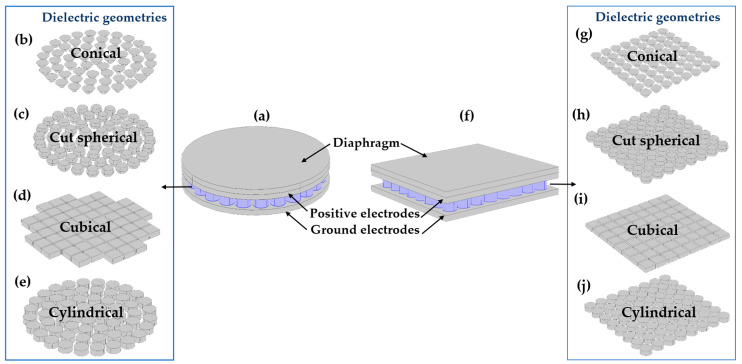
Proposed sensor geometry, (**a**) Geometry of the circular shape CPS. Geometries of the dielectric layer are arranged in a circular shape, (**b**) Conical shape, (**c**) Cut spherical shape, (**d**) Cubical shape, and (**e**) cylindrical shape. (**f**) Geometry of the cubical shape CPS. Geometries of the dielectric layer are arranged in a square shape (**g**) Conical, (**h**) cut spherical, (**i**) cubical, and (**j**) Cylindrical shape.

**Figure 4 sensors-24-03504-f004:**
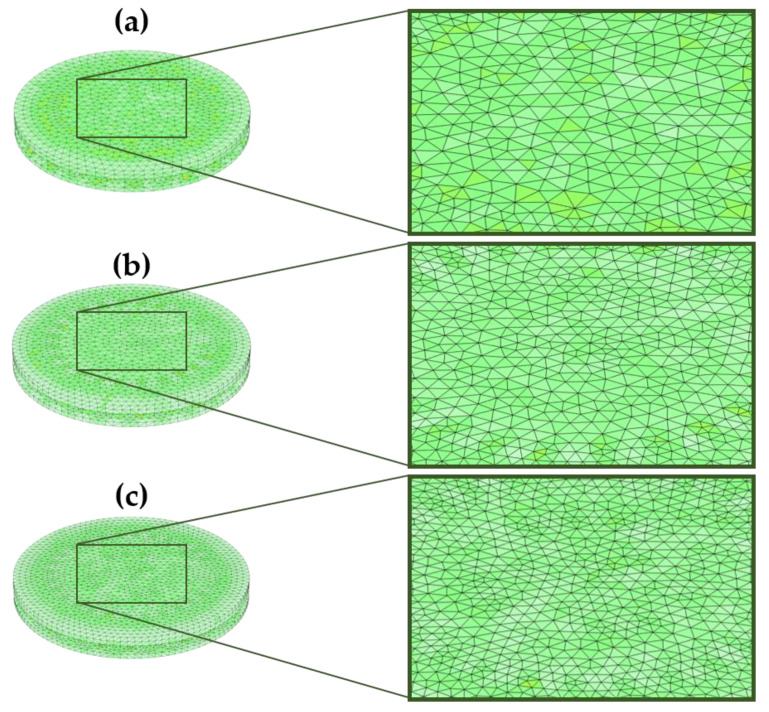
Meshing to the circular CPS consisting of cylindrical dielectric, (**a**) Normal mesh, (**b**) Fine mesh, and (**c**) Finer mesh.

**Figure 5 sensors-24-03504-f005:**
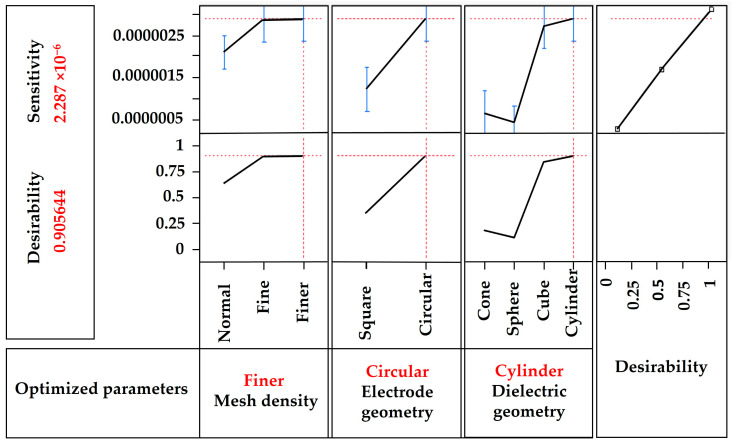
RSM optimization plot for the maximum sensitivity of CPS.

**Figure 6 sensors-24-03504-f006:**
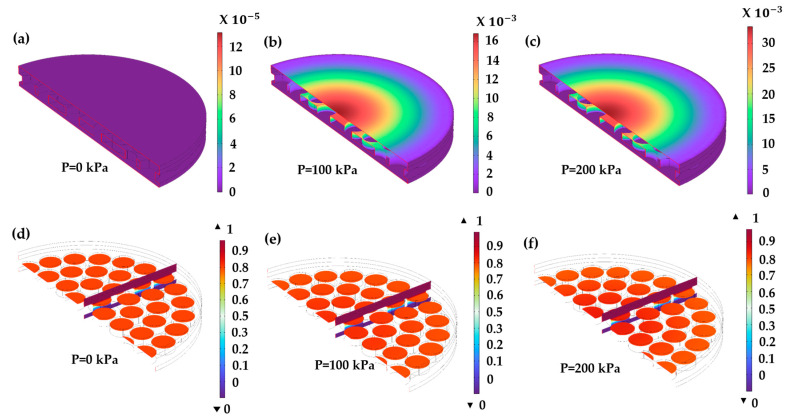
Contour plots to show the displacement of diaphragm and dielectric material and the electric potential in CPS. (**a**–**c**) displacement at 0, 100, and 200 kPa. (**d**–**f**) Electric potential distribution at 0, 100, and 200 kPa.

**Figure 7 sensors-24-03504-f007:**
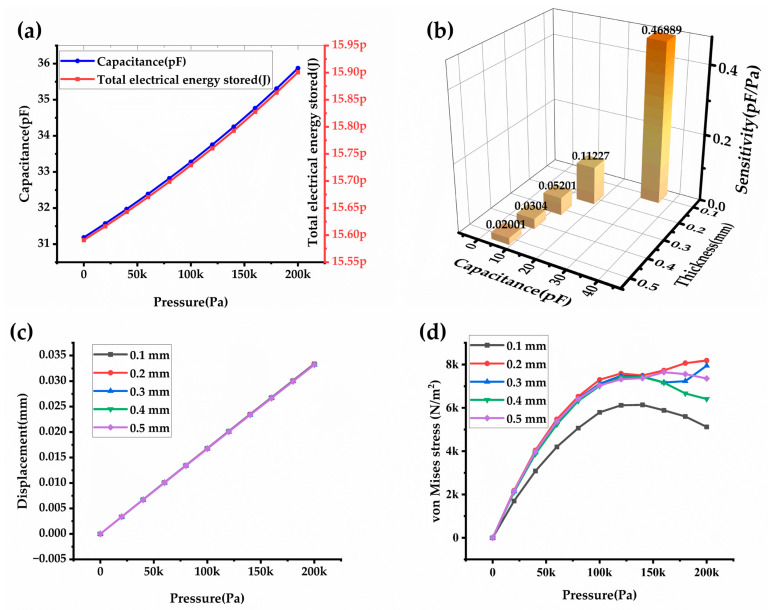
(**a**) Effect of pressure on the capacitance and total electrical energy stored for 0.1 mm dielectric thickness, (**b**) capacitive sensitivity characteristics, (**c**) Effect of pressure on the diaphragm displacement, and (**d**) von Mises stress analysis.

**Figure 8 sensors-24-03504-f008:**
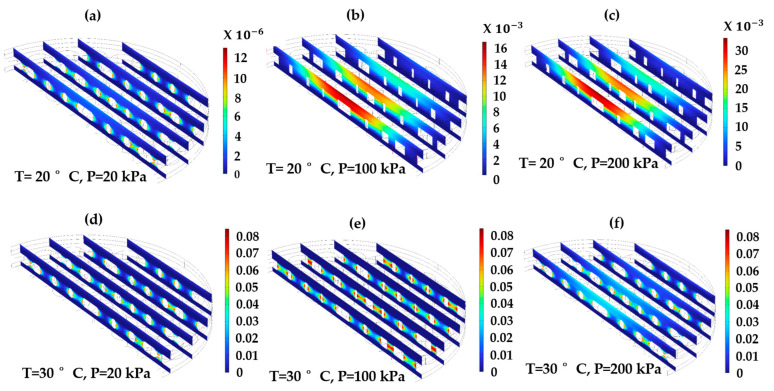
Sliced contour plot of the vertically half cross-section of the CPS to show the temperature effect. (**a**–**c**) displacement due to 20 °C at 20, 100, and 200 kPa. (**d**–**f**) displacement due to 30 °C at 20, 100, and 200 kPa.

**Figure 9 sensors-24-03504-f009:**
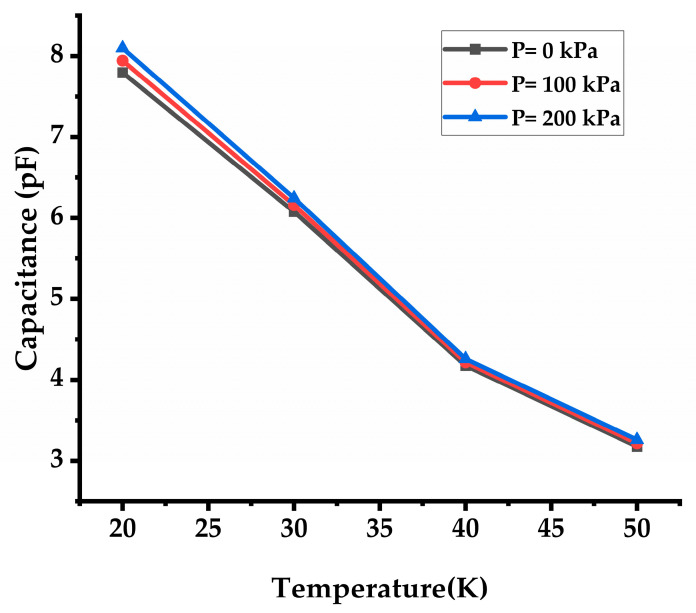
Variation in the capacitance due to the change in the operating temperature of the CPS.

**Figure 10 sensors-24-03504-f010:**
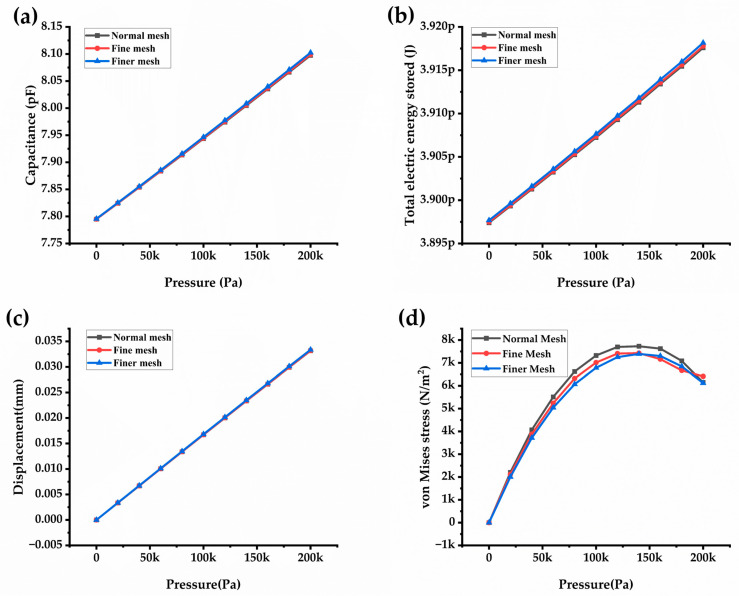
Mesh optimization for the electrical and mechanical parameters. (**a**) Capacitance against pressure, (**b**) Total electrical energy against pressure, (**c**) Displacement magnitude against pressure, and (**d**) von Mises stress against pressure.

**Table 1 sensors-24-03504-t001:** The material used for the CPS model and their respective properties.

Parameters	Materials	Dielectric Constant (ε)	Young’s Modulus(E) [GPa]	Poisson’s Ratio (ν)	Density (ρ) [kg/m^3^]
Diaphragm	PDMS	2.4–2.7	0.00075–0.00090	0.49	970
Dielectric	PVDF	10–12	2.17	0.40	1.78
Electrodes	Silver	0.188–1.971	83	0.37	10.8

**Table 2 sensors-24-03504-t002:** Electrode and diaphragm dimensions.

Square Electrode Setup	Circular Electrode Setup
Diaphragm/Electrodes	Diaphragm/Electrodes
Area-10 × 10 mm^2^	Radius—5 mm
Height—0.2 mm	Height—0.2 mm

**Table 3 sensors-24-03504-t003:** Dimension of the dielectric medium.

Conical	Cubical	Cut-Sphere	Cylindrical
Upper radius—0.5 mm	Width = 0.5 mm	Radius—0.5 mm (Cut on both sides and kept height 0.2 mm)	Radius—0.5 mm Height—Variable
Lower radius—0.3 mm	Length = 0.5 mm		
Height—0.2 mm	Hight = 0.2 mm		

**Table 4 sensors-24-03504-t004:** Factors and levels.

Factors	Description	Level 1	Level 2	Level 3	Level 4
X	Mesh density	0.1–1.5 mm	0.1–0.8 mm	0.04–0.55 mm	
		(normal)	(fine)	(finer)	-
Y	Dielectric geometry	Conical shape	Cut-spherical	Cubical	Cylindrical shape
Z	Diaphragm geometry/Electrode geometry	Square form	Circular form	-	-

**Table 5 sensors-24-03504-t005:** DoE table.

Treatments	Factor Assigned			Response (Sensitivity pF/Pa)
	Mesh Density	Dielectric Geometry	Electrode Geometry	
1	Normal	Cone	Square	0.9380
2	Finer	Cylinder	Circular	2.9945
3	Normal	Sphere	Circular	0.4040
4	Normal	Cube	Square	1.3683
5	Fine	Cylinder	Circular	2.9801
6	Fine	Sphere	Square	0.6848
7	Normal	Cone	Circular	0.5503
8	Fine	Sphere	Circular	0.4052
9	Finer	Cone	Circular	0.5611
10	Fine	Cone	Square	0.9531
11	Finer	Cube	Circular	2.7738
12	Normal	Cylinder	Square	1.1300
13	Finer	Cylinder	Square	1.1090
14	Fine	Cone	Circular	0.5578
15	Normal	Sphere	Square	0.6806
16	Normal	Cube	Circular	2.0147
17	Finer	Sphere	Circular	0.3962
18	Fine	Cylinder	Square	0.1108
19	Finer	Sphere	Circular	0.3962
20	Fine	Cube	Square	0.1343
21	Finer	Cone	Square	0.9624
22	Normal	Cylinder	Circular	2.4421
23	Fine	Cube	Square	1.3432
24	Fine	Cube	Circular	2.6648
25	Normal	Cone	Square	0.9380
26	Normal	Cylinder	Circular	1.5121
27	Finer	Cube	Square	1.3314
28	Finer	Sphere	Square	0.6680

**Table 6 sensors-24-03504-t006:** Regression analysis by linear model fitting.

Source	DF	Sum of Squares	Mean Square	F Ratio	Prob > F
Model	17	1.745 × 10^−11^	1.026 × 10^−12^	13.6232	<0.0001
Error	10	7.5349 × 10^−13^	7.535 × 10^−14^	-	-
Corrected total	27	1.8204 × 10^−11^	-	-	-

**Table 7 sensors-24-03504-t007:** Comparative study on the various geometry and materials in capacitive sensors.

Authors	Geometry	Dielectric Material	Electrode Material	Sensitivity	Application
Baek et al., 2017 [31]	Wrinkled dielectric layer	PDMS	Gold	4.8 × 10^−6^ to 5.2 × 10^−6^ kPa^−1^	Bio-medical devices
Choi et al., 2020 [32]	Porous composite	Porous eco flex-Multiwalled CNT	Conductive fabric	1.72 kPa^−1^	Electronic devices
Mahata et al., 2020 [33]	Scrubber composite	Rose petal-PMMA-double-layer PDMS	ITO/PET	0.055 kPa^−1^	e-skin
Parthasarathy and S, 2017 [34]	Flat square	Silicon/Silicon carbide	-	0.574 kPa^−1^	Altimeter
Ruth and Bao, 2020 [35]	Microstructure	PDMS	-	0.1 to 1 kPa^−1^	Robotics
Zhao et al., 2020 [36]	3D dielectric layer	Silver nanowire	Gold plated PDMS	1.21 kPa^−1^	Wearable electronics
Y. Kim et al., 2021 [37]	Cone shape dielectric	Porous PDMS	Aluminum	5 kPa^−1^	Sensor array and skin at touched sensors
S. Li et al., 2020 [14]	Super elastic dielectric	Porous PDMS Film	Conductive cloth tape	0.023 kPa^−1^	Detection of human motions
Bai et al., 2020 [28]	Graphene based Architecture	PVA/H_3_PO_4_ Ionic film	PI–Au	220 kPa^−1^ to 3300 kPa^−1^	e-skin
Xiong et al., 2020 [21]	Convex microarrays	PVDF	PDMS—PS—Au	30.2 kPa^−1^	Motion and health monitoring
S. W. Kim et al., 2022 [38]	Microarray	Barium titanate—PVDF	PDMS	4.9 kPa^−1^	e-skin and wearable medical assistive devices
S. W. Park et al., 2018 [39]	Elastomer dielectric	Porous PDMS	MWCNT/PEDOT: PSS	1.12 × 10^3^ kPa^−1^	Gait signal analysis
Wu et al., 2020 [40]	Microstructure dielectric	Spiky Ni/PDMS	Silver flakes/PDMS	0.0046 kPa^−1^	e-skin
Liu et al., 2021 [22]	Microstructure dielectric	Natural bamboo leaves	Ag NW/MXene	2.08 kPa^−1^	Breath/wrist pulse/speech, joint bending detecting sensors
Keum et al., 2021 [19]	Textile array	PVDF—HFP/IL	Ag-plated fibers on polyester	9.51 kPa^−1^	e-textile
Bijender and Kumar, 2020 [41]	Microstructure	Porous-PDMS scrubber composite	ITO/PET	0.0083 kPa^−1^	Wearable application
W. Li et al., 2020 [42]	Elastomer	Airgap-PDMS	PPy/Filter paper	0.0040 kPa^−1^	Wearable devices
Jung et al., 2020 [43]	3D	PDMS/MS Composite	ITO/PET	0.124 kPa^−1^	Detecting moments in fingers
Lei et al., 2014 [44]	Flat square	PDMS	indium-zinc-oxide	2.24 and 0.28 × 10^3^ Pa^−1^	Bio-Medical devices

## Data Availability

Data are contained within the article.

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
