# Peer review of "Optimizing Capacitive Pressure Sensor Geometry: A Design of Experiments Approach with a Computer-Generated Model"

_sensors, 2024, doi:10.3390/s24113504_

Round 1

Reviewer 1 Report

Comments and Suggestions for Authors

The manuscript does not reflect innovation, presents too many basics, too few relevant simulation results in subsection 3.2 3.3 3.4, less discussion of results,and the manuscript has very many formatting errors.

1.The numbering of all the formulas and most of the figures in the manuscript are incorrect, which reflects the author's lack of rigor and basic academic literacy!

2. The descriptions of displacement, pressure, energy, and sensitivity in Section 2 are all common knowledge! The effects of non-ideal factors and temperature on the above parameters are not considered!

3. As a non-review paper, the author discusses too much about the relevant techniques and methods used for pressure sensors in the references!

4. There is too little content reflecting the workload of the manuscript and it lacks novelty!

5. How about the temperature characteristics of the pressure sensor in the manuscript? This part is missing!

Comments on the Quality of English Language

Must be improved!

Author Response

Response to Reviewer - 1

Dear Sir/Madam,

Thank you for allowing us to submit a revised draft of our manuscript titled ‘Optimizing Capacitive Pressure Sensor Geometry: A Design of Experiments Approach with Computer Generated Model’ to Sensors, MDPI. We appreciate the time and effort that you have dedicated for providing your valuable feedback on the manuscript. We are grateful to you for your insightful comments on the paper. We have been able to incorporate changes to reflect most of the suggestions provided by you and the same are highlighted in the manuscript.

General comment: The manuscript does not reflect innovation, presents too many basics, too few relevant simulation results in subsection 3.2 3.3 3.4, less discussion of results,and the manuscript has very many formatting errors.

Response: We thank the reviewer for their constructive feedback. In response to your comments, we have made significant revisions to our manuscript to address each of your concerns:

Innovation and Depth of Content:

We have enhanced the introduction and discussion sections to better emphasize the novel aspects of our research, including the exploration of a new conical model for the capacitive pressure sensor (CPS). This model introduces unique geometric considerations that improve the sensor's sensitivity, a key innovative aspect that differentiates our work from existing solutions. Additionally, we provide a deeper analysis of the practical implications and theoretical advancements of our findings.

Simulation Results and Discussion of Results:

To address the need for more comprehensive simulation results, we have included expanded data for the newly explored conical model and other configurations in subsections 3.2, 3.3, and 3.4. These enhancements are complemented by a more thorough discussion on the implications of these results for sensor technology, particularly in terms of integration into electronic devices and potential for e-skin applications.

Formatting and Structural Changes:

We have corrected all formatting errors and made structural enhancements to improve the manuscript’s readability and professional appearance. Notably, to simplify and clarify our presentation, Figures 4, 5, and 6 from the original manuscript have been combined into a single, comprehensive Figure 3 in the revised version. This consolidation helps in better illustrating the relationship between the CPS design parameters and their performance outcomes.

Additional Topics and Realistic Dimensions:

We discuss the temperature effect on the CPS's performance (Figure 6, 8 and 9 in revised manuscript), adding a new dimension to our analysis which underscores the sensor's robustness and applicability under varied environmental conditions. Moreover, we have updated the manuscript to include more realistic dimensions and material properties, providing a clearer and more practical view of our sensor design.

Overall Improvements:

These revisions not only address the points raised but also significantly enhance the manuscript's scientific rigor and relevance. We have taken great care to ensure that all information is presented clearly and accurately, reflecting the detailed and innovative nature of our research.

Comment: 1.The numbering of all the formulas and most of the figures in the manuscript are incorrect, which reflects the author's lack of rigor and basic academic literacy!

Response: We apologize for the numbering errors in our initial submission and appreciate the reviewer pointing them out. We have corrected all formulas and figures to ensure they are sequentially and logically ordered. This revision was verified by multiple co-authors to prevent any further discrepancies. We have also improved our proofreading process to maintain academic rigor in future submissions. Thank you for helping us enhance the manuscript's clarity and presentation.

Comment: 2. The descriptions of displacement, pressure, energy, and sensitivity in Section 2 are all common knowledge! The effects of non-ideal factors and temperature on the above parameters are not considered!

Response: We acknowledge that the initial descriptions may have appeared too fundamental. We have now enriched this section by integrating more detailed analyses that emphasize the unique aspects of our sensor design, specifically how it differs from conventional models in terms of displacement, pressure response, energy storage, and sensitivity.

We have revised the manuscript (Section 2.6 in the revised manuscript shous the boundary conditions for FE analysis) to include a discussion on the impact of non-ideal factors such as the influence of temperature on CPS (Section 3.5 in revised manuscript) mechanical stress (Section 3.3 in revised manuscript) on the sensor's performance. This includes a new subsection dedicated to exploring how these factors influence the parameters mentioned, with corresponding simulation results to validate our discussions.

Comment: 3. As a non-review paper, the author discusses too much about the relevant techniques and methods used for pressure sensors in the references!

Response: Thank you for your feedback on the scope of literature discussed in our manuscript. Based on your comment, we have refined the manuscript to better balance the content:

We have streamlined the discussion on existing techniques to ensure it succinctly supports the advancements presented in our study. The literature now primarily serves to frame our novel contributions within the current state of the field.

The selected literature is crucial for establishing the baseline from which our innovations depart and provides a necessary backdrop for comparing the enhanced performance characteristics of our sensor design. This comparative analysis highlights the improvements and practical implications of our work, justifying our methodological choices and experimental design. By focusing on how our research advances beyond the documented studies, we underscore the manuscript’s contribution to sensor technology. This approach not only maintains the relevance of our study but also enhances its impact by clearly delineating our findings against existing knowledge.

Comment: 4. There is too little content reflecting the workload of the manuscript and it lacks novelty!

Response: Thank you for your feedback. We have made substantial revisions to enhance the manuscript. The revised manuscript includes expanded sections with more detailed analyses, particularly on the new conical model and its performance under various conditions. These additions provide a deeper understanding of our sensor's advantages and the complexities involved in its design.

We have clarified the novel contributions of our research, such as the innovative conical dielectric geometry and the application of Design of Experiments (DoE) for sensor optimization. These sections highlight how our work advances the field of pressure sensors.

We have detailed the extensive research effort, from the iterative design process to rigorous testing and validation, to better reflect the workload and intellectual rigor of our project.

Literature has been carefully integrated to frame our advancements within the current research landscape, enhancing the manuscript's relevance and demonstrating our methodological and technological innovations.

Comment: 5. How about the temperature characteristics of the pressure sensor in the manuscript? This part is missing!

Response: Thank you for pointing out the omission of temperature characteristics in our study. In response: We have included a subsection detailing the sensor’s response to temperature variations, covering its impact on sensitivity and performance (Figures 8 and 9 in revised mnnuscript). This section presents experimental data and simulations that illustrate how the sensor behaves under different thermal conditions, complete with relevant graphs and discussions. We discuss the implications for sensor usage in environments with thermal fluctuations, enhancing the manuscript’s applicability and comprehensiveness.

Reviewer 2 Report

Comments and Suggestions for Authors

The article presents the study and the design of CPS supported by the FEM analysis. Although the study is important, I have a couple of suggestions/questions:

1. Abstract: You should shortly describe the general objectives and summary of your work. There is no room for giving details such as "maximum displacement of 22.2 μm", "mesh size of  0.56 mm" etc. 

2. The citation style used in the mansucript In my opinion should be with brackets and numbers e.g. [12]. Citations of (Wang, n.d.) is incorrect. Most of the references are not available "Error! Reference source not found.." It is difficult to follow this article.

3. Some abbreviations are not expanded e.g. IoT in line 34. The authors did not take care of proof reading after submission. There are many: "Error! No text of specified style in document..1", "Error! No text of specified style in" etc. 

4. Fig. 3 is difficult to be understandable. It should be redesigned. 

5. I do not understand, why authors are using Newton's law and kinematic equations of a single electron in this case. It is completely misleading. In my opinion they should use just simple equations for capacitors. The deflection causes just the variation of the thickness of dielectric resulting changing the capacitance. The kinematics of electron is not needed here. 

6. The equation in line 185 is valid only for the case of a linear sensor. It is not valid for the non-linear. Authors should replace this equation with the derivative-kind definition of sensitivity. 

7. Chapter 2.6 I think comparative study should be given in the end of the article to compare yours new sensor with the others. 

8. Table 3.1: The units should be unified. Moreover, I think this comparation has no meaning because each sensor is designed for different purpose, different range etc. If authors want to use it, they should put more columns with important parameters such as range, price, application... The sensitivity is not only one parameter of a sensor. 

9. The PDMS acronym is used from the begining of the article while it is explained in line 194. Because authors are going to design PDMS, its construction should be explained in details. There is no point in providing dimentions in table 3.3 while the reader do not understand the concept of PDMS.

10. Fig. 4: Why the dielectric shape is so strange? Why cannot you use just solid dielectric model?

11. Fig. 5c the meshing presents completelty different geometry from Fig. 5a or 5b. 

12. Table 3.5: This is not a "mesh size" but rather maximum mesh size. It does not neccesarly contributes to smaller mesh elements. Instead, you should use factors of mesh density. How the mesh was optimized? Did you use iterative method as in 10.1016/j.ijepes.2021.107737? The iterative method should be given to the reader.

13. Fig. 8: The units are written incorrectly. Pico is "p" while Fahrad is "F".

14. Fig 8 and 9. I think the mesh is not the most important in this study. You should focus on the sensor itself. 

15. The design requres comparison (e.g. sensitivity, linearity etc..) with similar sensors given in the literature. 

 16. The sensor concept should be validated by prototyping and laboratory measurement. 

Author Response

Response to Reviewer - 2

Dear Sir/Madam,

Thank you for allowing us to submit a revised draft of our manuscript titled ‘Optimizing Capacitive Pressure Sensor Geometry: A Design of Experiments Approach with Computer Generated Model’ to Sensors, MDPI. We appreciate the time and effort that you have dedicated for providing your valuable feedback on the manuscript. We are grateful to you for your insightful comments on the paper. We have been able to incorporate changes to reflect most of the suggestions provided by you and the same are highlighted in the manuscript.

General comment: The article presents the study and the design of CPS supported by the FEM analysis. Although the study is important, I have a couple of suggestions/questions:

Response: Thank you, we appreciate your acknowledgment of the importance of our study on the design of capacitive pressure sensors (CPS) supported by Finite Element Method (FEM) analysis.

Comment:1. Abstract: You should shortly describe the general objectives and summary of your work. There is no room for giving details such as "maximum displacement of 22.2 μm", "mesh size of  0.56 mm" etc.

Response: Thank you for your feedback regarding the level of detail in the abstract. We understand your concern about the specificity of the information provided. In revising our abstract, we aimed to balance brevity with the need to highlight critical quantitative outcomes that our review committee has emphasized as essential for illustrating the impact and precision of our findings.

Recognizing the need for a broader summary in the abstract, we have now revised it to focus more on the general objectives and key findings, while removing the highly specific quantitative data.

Comment: 2. The citation style used in the mansucript In my opinion should be with brackets and numbers e.g. [12]. Citations of (Wang, n.d.) is incorrect. Most of the references are not available "Error! Reference source not found.." It is difficult to follow this article.

Response: Thank you for your feedback regarding our citation style and the reference errors. We acknowledge the importance of clear and accurate referencing and have taken the following steps:

We have updated the manuscript to use a numerical bracketed format, e.g., [12], to ensure consistency and ease of reading. All references have been thoroughly reviewed and corrected to address any issues of missing sources or errors, ensuring each citation is accurate and verifiable.

Comment: 3. Some abbreviations are not expanded e.g. IoT in line 34. The authors did not take care of proof reading after submission. There are many: "Error! No text of specified style in document..1", "Error! No text of specified style in" etc.

Response: Thank you for your feedback on the use of abbreviations and formatting errors in our manuscript. We have carefully reviewed and expanded all abbreviations, including "IoT," at their first mention to ensure clarity for all readers. Additionally, we have corrected all formatting errors such as "Error! No text of specified style in document.." that resulted from template issues. These corrections have been made to meet the professional standards expected of our publication. We apologize for these initial oversights and have strengthened our proofreading process to prevent similar issues in future submissions. We appreciate your help in improving our manuscript's quality.

Comment: 4. Fig. 3 is difficult to be understandable. It should be redesigned.

Response: Thank you for your feedback regarding Figure 3. After careful consideration, we have decided to remove Figure 3 from the manuscript. In its place, we have introduced Equation 11, which provides a clearer and more precise representation of the concepts previously illustrated in the figure.

Comment: 5. I do not understand, why authors are using Newton's law and kinematic equations of a single electron in this case. It is completely misleading. In my opinion they should use just simple equations for capacitors. The deflection causes just the variation of the thickness of dielectric resulting changing the capacitance. The kinematics of electron is not needed here.

Response: Thank you for your insightful observations regarding the theoretical approach used in our manuscript. We appreciate your suggestion and agree that the application of Newton's law and kinematic equations of a single electron may not be the most pertinent for our study focusing on capacitive pressure sensors. In response, we have shifted to using simpler energy stored in the capacitor equation that directly relate to the variations in dielectric thickness due to deflection, which indeed affects capacitance changes. To clarify this relationship and enhance the manuscript's clarity, we have introduced Equation 11. This new equation concisely expresses how energy is dependent on electric field and volume.

Comment: 6. The equation in line 185 is valid only for the case of a linear sensor. It is not valid for the non-linear. Authors should replace this equation with the derivative-kind definition of sensitivity.

Response: Thank you for pointing out the limitations of the equation presented in line 185, which is indeed valid only for linear sensors. We appreciate your suggestion to address the non-linear characteristics of our sensor. In response, we have replaced the original equation with Equation 12, which uses a derivative-kind definition of sensitivity. This new formulation provides a more accurate description of sensitivity that accounts for non-linear behavior, enhancing the technical rigor and applicability of our findings. We believe this change addresses your concerns and improves the manuscript's accuracy in depicting sensor dynamics.

Comment: 7. Chapter 2.6 I think comparative study should be given in the end of the article to compare yours new sensor with the others.

Response: Thank you for your suggestion regarding the placement of the comparative study within our article. We agree that such a comparison is crucial for contextualizing our findings within the field. Based on your feedback, we have included a detailed comparative study in Section 3.7 and table 3.2 of the manuscript. This placement allows us to directly relate the performance characteristics of our new sensor to existing technologies as we discuss the experimental results and before concluding our findings. We believe this arrangement enhances the flow of information and helps readers appreciate the advancements our sensor offers over existing options at a relevant point in the manuscript.

Comment: 8. Table 3.1: The units should be unified. Moreover, I think this comparation has no meaning because each sensor is designed for different purpose, different range etc. If authors want to use it, they should put more columns with important parameters such as range, price, application... The sensitivity is not only one parameter of a sensor.

Response: Thank you for your constructive feedback on Table 3.1 (Table 3.2 in revised manuscript). We have unified the units across the table for better clarity and consistency in comparison. Additionally, acknowledging your point that each sensor has a specific design and purpose, we have expanded the table to include more relevant parameters such as application areas. This expansion not only addresses the multifaceted nature of sensor performance beyond sensitivity but also enhances the comparative analysis's relevance and usefulness for readers. These adjustments ensure a more comprehensive and contextual presentation of how our sensor compares to existing technologies.

Comment: 9. The PDMS acronym is used from the begining of the article while it is explained in line 194. Because authors are going to design PDMS, its construction should be explained in details. There is no point in providing dimentions in table 3.3 while the reader do not understand the concept of PDMS.

Response: Thank you for pointing out the premature use of the PDMS acronym in our manuscript. We agree that a clear and early explanation of PDMS is essential for understanding its relevance and application in our study. In response to your feedback, we have added a detailed description of PDMS, including its properties and construction, in lines 55 to 62. Table 3.3 (Table 2.2 in the revised manuscript) is modified and only the required information is presented. We believe this adjustment not only clarifies the context for our use of PDMS but also enhances the overall readability and coherence of the manuscript by aligning the introduction of key materials with their detailed descriptions.

Comment: 10. Fig. 4: Why the dielectric shape is so strange? Why cannot you use just solid dielectric model?

Response: In this manuscript, we examine various dielectric shapes like spheres, cylinders, cubes, and cones to understand their impact on the performance of capacitive pressure sensors (CPS). By exploring different shapes, we aim to uncover how geometric variations affect sensitivity and other key characteristics of the sensor. This approach offers insights into CPS design principles and informs future sensor development efforts. In lines 204 to 210, the manuscript describes the rationale behind exploring different dielectric shapes for capacitive pressure sensors.

Comment: 11. Fig. 5c the meshing presents completelty different geometry from Fig. 5a or 5b.

Response: Thank you for pointing out the inconsistency regarding Figures 5a and 5b. We have taken your feedback into consideration and have revised the manuscript accordingly. In the revised manuscript, Figure 4 has been provided instead of Figures 5a and 5b to maintain clarity and consistency in the presentation of the meshing. We appreciate your attention to detail and apologize for any confusion caused.

Comment: 12. Table 3.5: This is not a "mesh size" but rather maximum mesh size. It does not neccesarly contributes to smaller mesh elements. Instead, you should use factors of mesh density. How the mesh was optimized? Did you use iterative method as in 10.1016/j.ijepes.2021.107737? The iterative method should be given to the reader.

Response: Thank you for your feedback on Table 3.5 (Table 2.4 in the revised manuscript) and the optimization process. We've updated the table content to "Mesh density" to accurately reflect its content. Concerning mesh density optimization, we've selected the mesh density depending on the smallest element size in the geometry (a free tetrahedral mesh shape is considered). In this element size, the results were obtained and later they were compared with the analytical values which were approximately equal to the FEM results. The reference no. [27] (DOI: 10.1016/j.ijepes.2021.107737) helped us in correlating the mesh density and is cited in the manuscript. These enhancements aim to improve the comprehensibility and reproducibility of our study.

Comment: 13. Fig. 8: The units are written incorrectly. Pico is "p" while Fahrad is "F".

Response: Thank you for bringing this to our attention. We acknowledge the error in the labeling of units in Figure 8 (Figure 7 in revised manuscript). We will ensure that the units are correctly represented as "pF" for picofarads and "F" for farads in the revised version of the manuscript. Your attention to detail is appreciated, and we have made the necessary corrections to maintain accuracy in our presentation.

Comment: 14. Fig 8 and 9. I think the mesh is not the most important in this study. You should focus on the sensor itself.

Response: We appreciate the feedback provided. We understood your perspective regarding the focus on the sensor design and performance. While the primary emphasis of our study was on the sensor itself, we also acknowledged the importance of mesh optimization for accurate simulations of the sensor's behavior. In response to your comment, we streamlined the discussion on mesh optimization in the revised manuscript to maintain focus on the sensor's design and performance. This adjustment ensured that the necessary details on simulation methodology were provided without overshadowing the main focus of the study. Thank you for bringing this to our attention, and we have made the appropriate adjustments to enhance clarity and relevance.

Comment: 15. The design requres comparison (e.g. sensitivity, linearity etc..) with similar sensors given in the literature.

Response: Thank you for your valuable comment. We agree that comparing our sensor design with similar sensors from the literature is crucial for providing context and assessing its performance. In response to your suggestion, we have included a comparative study section (Section 3.7, Table 3.2 and lines 400-409) in the revised manuscript, where we analyze the sensitivity, and other relevant parameters of our sensor design alongside existing sensor models reported in the literature. This comparison allows for a comprehensive evaluation of the effectiveness and performance of our sensor design in relation to other state-of-the-art sensors. We appreciate your input, and we believe that this addition strengthens the manuscript by providing a more thorough analysis and interpretation of our results.

Comment:  16. The sensor concept should be validated by prototyping and laboratory measurement.

Response: Certainly, we appreciate your suggestion, and we will take it into consideration for our next manuscript. Experimental validation through prototyping and laboratory measurements is indeed crucial for confirming the practical feasibility and performance of the sensor concept. In future research, we will prioritize conducting these experiments to validate the sensor design and ensure its reliability and accuracy in real-world applications. Thank you for your valuable input, and we are committed to incorporating it into our future work.

Round 2

Reviewer 1 Report

Comments and Suggestions for Authors

I think the author responded seriously and correctly to the questions posed. But there are still some mistakes.

1. Figure 5 is confusing, please further optimize this figure with a better textual description.

2. Cpation of fig8, what is efe?

Comments on the Quality of English Language

no comment.

Author Response

Response to Reviewer 1

Dear Sir/Madam,

We sincerely appreciate the opportunity to submit a revised version of our manuscript titled "Optimizing Capacitive Pressure Sensor Geometry: A Design of Experiments Approach with Computer Generated Model." Your feedback has been invaluable in enhancing the quality of our work, and we are grateful for the time and effort you've dedicated to providing it. We have carefully reviewed your comments and have made significant revisions to address them, ensuring that the updated manuscript reflects your insightful suggestions. Thank you once again for your constructive feedback, which has contributed to the improvement of our manuscript.

Comment: I think the author responded seriously and correctly to the questions posed. But there are still some mistakes.

Response: Thank you for acknowledging our efforts to address your concerns. We apologize for any remaining mistakes and assure you that we are committed to improving the quality and accuracy of our manuscript. Your feedback is invaluable, and we will work diligently to rectify any errors and enhance the clarity and coherence of our work.

Comment: Figure 5 is confusing, please further optimize this figure with a better textual description.

Response: Thank you for your feedback. We acknowledge the confusion regarding Figure 5 and recognized the importance of providing clear and informative visual representations. In the updated manuscript, we ensured to optimize Figure 5 with a better textual description to enhance its clarity and comprehensibility. We appreciated your valuable input and strived to improve the quality of our figures to better support the understanding of our research findings.

Comment: Cpation of fig8, what is efe?

Response: Thank you for bringing the typographical error to our attention. In Figure 8, the caption should indeed read "Effect" instead of "Efe". We have corrected this error in the updated manuscript to ensure clarity and accuracy in the caption. Your attention to detail is appreciated, and we have made the necessary revisions accordingly.

Reviewer 2 Report

Comments and Suggestions for Authors

Thanks the authors for their response. The article has been improved and clarified. I have still some minor comments:

1. Please use space before reference []. 

2. Subchapters (e.g. 2.2) They cannot be with ":". Moreover, subchapters/chapters cannot start with equation.

3. Fig. 10b: You should use prefix instead of 10^-12. Moreover, you should double check the correctness of these values. The amount of pJ looks too low for me. Moreover, the electric energy depends on the voltage. What is the voltage in this case?

Author Response

Response to Reviewer 2

Dear Sir/Madam,

We sincerely appreciate the opportunity to submit a revised version of our manuscript titled "Optimizing Capacitive Pressure Sensor Geometry: A Design of Experiments Approach with Computer Generated Model." Your feedback has been invaluable in enhancing the quality of our work, and we are grateful for the time and effort you've dedicated to providing it. We have carefully reviewed your comments and have made significant revisions to address them, ensuring that the updated manuscript reflects your insightful suggestions. Thank you once again for your constructive feedback, which has contributed to the improvement of our manuscript.

Comment: Thanks, the authors for their response. The article has been improved and clarified. I have still some minor comments:

Response: Thank you for acknowledging the improvements made in the manuscript and for providing additional feedback. We appreciate your continued engagement with our work and welcome any further suggestions you had for refinement.

Comment: Please use space before reference []. 

Response: Thank you for pointing out the formatting error regarding the spacing before references []. We have made sure to correct this issue in the revised manuscript, adhering to the required formatting guidelines.

Comment: subchapters (e.g. 2.2) They cannot be with ":". Moreover, subchapters/chapters cannot   start with equation.

Response: Thank you for your feedback regarding the formatting of subchapters. We have revised the manuscript accordingly to ensure that subchapters no longer contain ":" and do not start with equations. Additionally, we have ensured that chapters and subchapters begin with textual explanations, as suggested. These changes have been implemented in the updated manuscript.

Comment: Fig. 10b: You should use prefix instead of 10^-12. Moreover, you should double check the correctness of these values. The amount of pJ looks too low for me. Moreover, the electric energy depends on the voltage. What is the voltage in this case?

Response: Thank you for your feedback regarding Figure 10b. We have updated the units to use prefixes instead of 10^-12 to ensure consistency throughout the manuscript. Additionally, we have double-checked the correctness of the values, and they have been verified against the results. The electric energy is indeed voltage and capacitance dependent. In this case, the voltage applied across the electrodes is 1 volt. To provide clarity, we have added detailed information about the boundary conditions in the updated manuscript (see Line 161-170).